# On the Transfer of Inductive Bias from Simulation to the Real World: a New Disentanglement Dataset

**Muhammad Waleed Gondal**[1][*][†]    **Manuel Wüthrich**[1][*]    **Đorđe Miladinović**[2]
**Francesco Locatello**[12]    **Martin Breidt**[3]    **Valentin Volchkov**[1]    **Joel Akpo**[1]
**Olivier Bachem**[4]    **Bernhard Schölkopf**[1]    **Stefan Bauer**[1][†]

[1]Max Planck Institute for Intelligent Systems
[2]Department of Computer Science ETH Zurich
[3]Max Planck Institute for Biological Cybernetics
[4]Google Research, Brain Team

## Abstract

Learning meaningful and compact representations with disentangled semantic aspects is considered to be of key importance in representation learning. Since real-world data is notoriously costly to collect, many recent state-of-the-art disentanglement models have heavily relied on synthetic toy data-sets. In this paper, we propose a novel data-set which consists of over one million images of physical 3D objects with seven factors of variation, such as object color, shape, size and position. In order to be able to control all the factors of variation precisely, we built an experimental platform where the objects are being moved by a robotic arm. In addition, we provide two more datasets which consist of simulations of the experimental setup. These datasets provide for the first time the possibility to systematically investigate how well different disentanglement methods perform on real data in comparison to simulation, and how simulated data can be leveraged to build better representations of the real world. We provide a first experimental study of these questions and our results indicate that learned models transfer poorly, but that model and hyperparameter selection is an effective means of transferring information to the real world.

## 1   Introduction

In representation learning it is commonly assumed that a high-dimensional observation $\mathbf{X}$ is generated from low-dimensional factors of variation $\mathbf{G}$. The goal is usually to revert this process by searching for a latent embedding $\mathbf{Z}$ which replicates the underlying generative factors $\mathbf{G}$, e.g. shape, size or color. Learning well-*disentangled* representations of complex sensory data has been identified as one of the key challenges in the quest for artificial intelligence (AI) [2, 45, 31, 3, 48, 29, 54], since they should contain all the information present in the observations in a compact and interpretable structure [2, 26, 8] while being independent from the task at hand [15, 33].

Disentangled representations may be useful for (semi-)supervised learning of downstream tasks, transfer and few-shot learning [2, 49, 39]. Further, such representations allow to filter out nuisance factors [27], to perform interventions and to answer counterfactual questions [44, 50, 45]. First

---

[*]These authors contributed equally.
[†]Correspondence to: `waleed.gondal@tue.mpg.de`, `stefan.bauer@inf.ethz.ch`

applications of algorithms for learning disentangled representations have been applied to visual concept learning, sequence modeling, curiosity-based exploration or even domain adaptation in reinforcement learning [51, 30, 42, 20, 22, 34, 54]. The research community is in general agreement on the importance of this paradigm and much progress has been made in the past years, particularly on the algorithmic level [e.g. 18, 24], fundamental understanding [e.g. 17, 52] and experimental evaluation [38]. However, research has thus far focused on synthetic toy datasets.

The main motivation for using synthetic datasets is that they are cheap, easy to generate and the independent generative factors can be easily controlled. However, real-world recordings exhibit *imperfections* such as chromatic aberrations in cameras and complex surface properties of objects (e.g. reflectance, radiance and irradiance), making transfer learning from synthetic to real data a nontrivial task. Despite the growing importance of the field and the potential societal impact in the medical domain or fair decision making [e.g. 6, 10, 37], the performance of state-of-the-art disentanglement learning on real-world data is unknown.

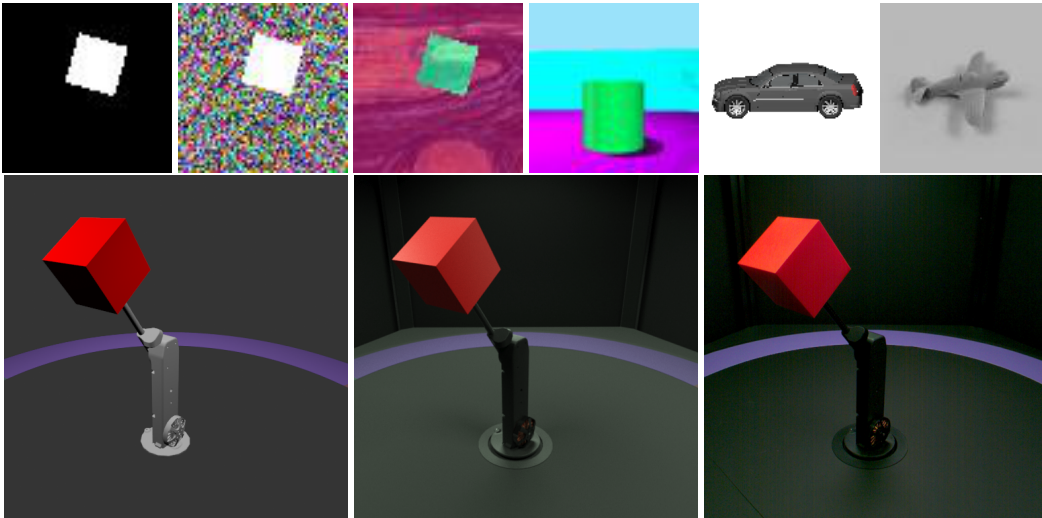

Figure 1: **Datasets:** In the top row samples from previously published datasets are shown from left to right: dSprites, Noisy-dSprites, Scream-dSprites, 3dshapes, Cars3D and SmallNORB. In the second row (again left to right) we provide simple simulated data, highly realistically simulated data and real-world data examples of the newly collected dataset.

To bridge the gap between simulation and the physical world, we built a recording platform which allows to investigate the following research questions: (i) How well do unsupervised state-of-the-art algorithms transfer from rendered images to physical recordings? (ii) How much does this transfer depend on the quality of the simulation? (iii) Can we learn representations on low dimensional recordings and transfer them from the current state-of-the-art of $64 \times 64$ images to high quality images? (iv) How much supervision is necessary to encode the necessary inductive biases? (v) Are the confounding and distortions of real-world recordings beneficial for learning disentangled representations? (vi) Can we disentangle causal mechanisms [44, 28, 29, 45] in the data generating process? (vii) Are disentangled representations useful for solving the real-world downstream tasks?

While answering all of the above questions is beyond the scope of this paper, our key contributions can be summarized as follows:

- We introduce the first *real-world 3D data set* recorded in a controlled environment, defined by *7 factors of variation*: object color, object shape, object size, camera height, background color and two degrees of freedom of motion of a robotic arm. The dataset is made publicly available[3].
- We provide synthetic images produced by computer graphics with two levels of realism. Since the robot arm and the objects are printed from a 3D template, we can ensure a close similarity between the realistic renderings and the real-world recordings.

- The collected dattset of physical 3D objects consists of over one million images, and each of the two simulated datasets contains the same number of images as well.
- We investigate the role of inductive bias and the transfer of different hyper-parameter settings between the different simulations and the real-world and the requirements on the quality of the simulation for a succesful transfer.

## 2 Background and Related Work

We assume a set of observations of a (potentially high dimensional) random variable $\boldsymbol{X}$ which is generated by $K$ unobserved causes of variation (generative factors) $\boldsymbol{G} = [G_1, \ldots, G_K]$ (i.e., $\boldsymbol{G} \rightarrow \boldsymbol{X}$) that do not cause each other. These latent factors represent *elementary ingredients* to the causal mechanism generating $\boldsymbol{X}$ [44, 45]. The elementary ingredients $G_i, i = 1, \ldots, K$ of the causal process work on their own and are changeable without affecting others, reflecting the independent mechanisms assumption [49]. However, for some of the factors a hierarchical structure may exist for which this may only hold true when seeing the hierachical structure as a whole as one component. The graphical model corresponding to this framework and adapted from [52] is depicted in figure 2. The hirachical structure of the factors $G_{K-1}^1$ and $G_{K-1}$ might represent one compositional process e.g. connected joints of a robot arm. The most commonly accepted understanding of *disentanglement* [2] is that each learned feature in $\boldsymbol{Z}$ should capture one factor of variation in $\boldsymbol{G}$.

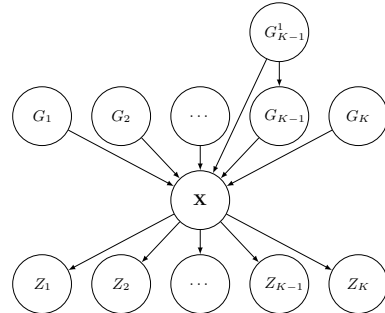

Figure 2: Graphical Model, where $\boldsymbol{G} = (G_1, G_2, \ldots, G_K)$ are the generative factors (color, shape, size, ...) and $\boldsymbol{X}$ the recorded images. The aim of disentangled representation learning is to learn variables $Z_i$ that capture the independent mechanisms $G_i$.

Current state-of-the-art disentanglement approaches use the framework of variational auto-encoders (VAEs) [25]. The (high-dimensional) observations $\boldsymbol{x}$ are modelled as being generated from some latent features $\boldsymbol{z}$ with chosen prior $p(\boldsymbol{z})$ according to the probabilistic model $p_\theta(\boldsymbol{x}|\boldsymbol{z})p(\boldsymbol{z})$. The generative model $p_\theta(\boldsymbol{x}|\boldsymbol{z})$ as well as the proxy posterior $q_\phi(\boldsymbol{z}|\boldsymbol{x})$ can be represented by neural networks, which are optimized by maximizing the variational lower bound (ELBO) of $\log p(\boldsymbol{x}_1, \ldots, \boldsymbol{x}_N)$

$$\mathcal{L}_{VAE} = \sum_{i=1}^{N} \mathbb{E}_{q_\phi(\boldsymbol{z}|\boldsymbol{x}^{(i)})}[\log p_\theta(\boldsymbol{x}^{(i)}|\boldsymbol{z})] - D_{KL}(q_\phi(\boldsymbol{z}|\boldsymbol{x}^{(i)})\|p(\boldsymbol{z}))$$

Since the above objective does not enforce any structure on the latent space, except for some similarity to the typically chosen isotropic Gaussian prior $p(\boldsymbol{z})$, various proposals for more structure imposing regularization have been made. Using some sort of supervision [e.g. 43, 4, 35, 40, 9] or proposing completely unsupervised [e.g. 19, 24, 7, 27, 13] learning approaches. [19] proposed the $\beta$-VAE penalizing the Kullback-Leibler divergence (KL) term in the VAE objective more strongly, which encourages similarity to the factorized prior distribution. Others used techniques to encourage statistical independence between the different components in $\boldsymbol{Z}$, e.g., FactorVAE [24] or $\beta$-TCVAE [7], while DIP-VAE proposed to encourage factorization of the inferred prior $q_\phi(\boldsymbol{z}) = \int q_\phi(\boldsymbol{z}|\boldsymbol{x})p(\boldsymbol{x}) \, d\boldsymbol{x}$. For other related work we refer to the detailed descriptions in the recent empirical study [38].

### 2.1 Established Datasets for the Unsupervised Learning of Disentangled Representations

Real-world data is costly to generate and groundtruth is often not available since significant confounding may exist. To bypass this limitation, many recent state-of-the-art disentanglement models [55, 24, 7, 18, 8] have heavily relied on synthetic toy datasets, trying to solve a simplified version of the problem in the hope that the conclusions drawn might likewise be valid for real-world settings. A quantitative summary of the most widely used datasets for learning disentangled representations is provided in table 1.

**Dataset Descriptions:** For quantitative analysis, *dSprites*[4] is the most commonly used dataset. This synthetic dataset [18] contains binary 2D images of hearts, ellipses and squares in low resolution.

| Dataset | Factors of Variation | Resolution | # of Images | 3D | Real-World |
|---|---|---|---|---|---|
| dSprites | 5 | $64 \times 64$ | 737,280 | ✗ | ✗ |
| Noisy dspirtes | 5 | $64 \times 64$ | 737,280 | ✗ | ✗ |
| Scream dSprites | 5 | $64 \times 64$ | 737,280 | ✗ | ✗ |
| SmallNORB | 5 | $128 \times 128$ | 48,600 | ✓ | ✗ |
| Cars3D | 3 | $64 \times 64$ | 17,568 | ✓ | ✗ |
| 3dshapes | 6 | $64 \times 64$ | 480,000 | ✓ | ✗ |
| *MPI3D-toy* | 7 | $64 \times 64$ | 1,036,800 | ✓ | ✗ |
| *MPI3D-realistic* | 7 | $256 \times 256$ | 1,036,800 | ✓ | ✗ |
| *MPI3D-real* | 7 | $512 \times 512$ | 1,036,800 | ✓ | ✓ |

Table 1: Summary of the properties of different datasets. The newly contributed datasets are *emphasized*.

In *Color-dSprites* the shapes are colored with a random color, *Noisy-dSprites* considers white-colored shapes on a noisy background and in *Scream-dSprites* the background is replaced with a random patch in a random color shade extracted from the famous The Scream painting [41]. The dSprites shape is embedded into the image by inverting the color of its pixel. The *SmallNORB*[5] dataset contains images of 50 toys belonging to 5 generic categories: four-legged animals, human figures, airplanes, trucks, and cars. The objects were imaged by two cameras under 6 lighting conditions, 9 elevations (30 to 70 degrees every 5 degrees), and 18 azimuths (0 to 340 every 20 degrees) [32]. For *Cars3D*[6],199 CAD models from [14] were used to generate 64x64 color renderings from 24 rotation angles each offset by 15 degrees [46]. Recently, *3dshapes* was made publicly available[7], a dataset of 3D shapes procedurally generated from 6 ground truth independent latent factors. These factors are floor colour, wall colour, object colour, scale, shape and orientation [24].

## 3 Bridging the Gap Between Simulation and the Real World: A Novel Dataset

While other real-world recordings, e.g. *CelebA* [36], exist, they offer only qualitative evaluations. However, a more controlled dataset is needed to quantitatively investigate the effects of inductive biases, sample complexity and the interplay of simulations and the real-world.

### 3.1 Controlled Recording Setup

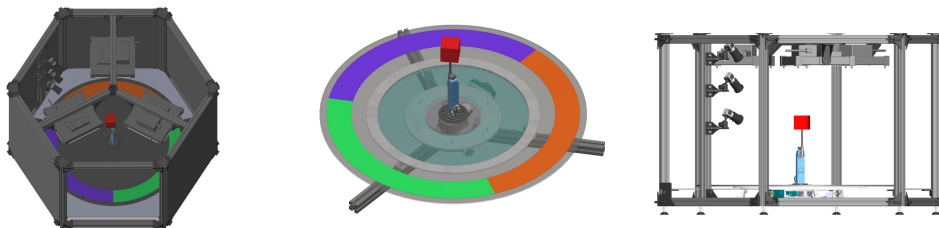

Figure 3: Renderings of the developed robotic platform: On the left a view from a 30° angle from the top (note that one panel in front and the top panels have been removed such that the interior of the platform is visible. During recordings, the platform is entirely closed. Middle: the robotic arm carrying a red cube (the entire cage is hidden). Right: frontal view without the black shielding (note the three cameras at different heights).

In order to record a controlled dataset of physical 3D objects, we built the mechanical platform illustrated in figure 3. It consists of three cameras mounted at different heights, a robotic manipulator carrying a 3D printed object (which can be swapped) in the center of the platform and a rotating table at the bottom. The platform is shielded with black sheets from all sides to avoid any intrusion of external factors (e.g. light) and the whole environment is relatively uniformly illuminated by three light sources installed within the platform.

### 3.1.1 Factors of Variation

The generative factors of variation $G$ mentioned in section 2 are listed in the following for our recording setup.

**Object Color:** All objects have one of six different colors: red (255, 0, 0), green (0, 255, 0), blue (0, 0, 255), white (255, 255, 255), olive (210,210,80) and brown (153,76,0) (see figure 4).

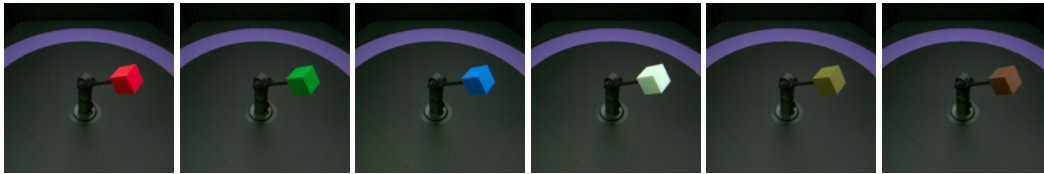

Figure 4: We show all the object colors while maintaining the other factors constant.

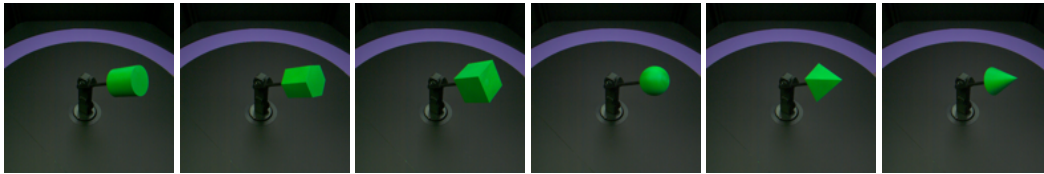

Figure 5: We show all object shapes while maintaining all other factors constant.

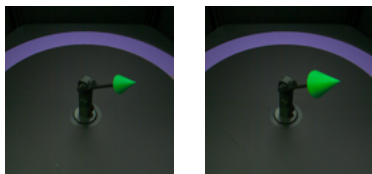

Figure 6: We show the two object sizes while maintaining all other factors constant.

**Object Shape:** There are objects of four different shapes in the dataset: a cylinder, a hexagonal prism, a cube, a sphere, a pyramid with square base and a cone. All objects exhibit rotational symmetries about some axes, however the kinematics of the robot are such that these axes never align with the degrees of freedom of the robot. This is important because it ensures that the robot degrees of freedom are observable given the images.

**Object Size:** There are objects of two different sizes in the dataset, categorized as large (roughly 65mm in diameter) and small (roughly 45 mm in diameter).

**Camera Height:** The dataset is recorded with three cameras mounted at three different heights (see figure 7 on the right), which represents another factor of variation in the images.

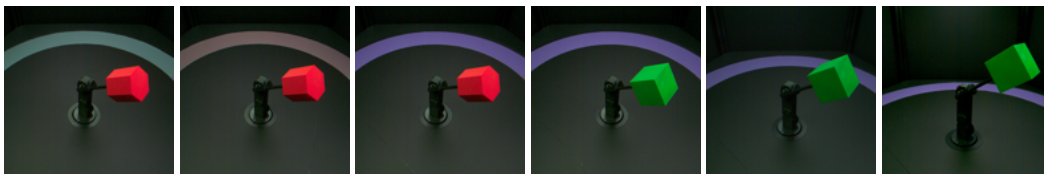

Figure 7: Three images on the left: we vary the backround color. Three images on the right: we vary the camera height.

**Background Color:** The rotation table (see figure 7) allows us to change background color. Note that for all images in the dataset we orient the table in such a way that only one background color is visible at a time. The colors are: sea green, salmon and purple.

**Degrees of Freedom of the Robotic Arm:** Each object is mounted on the tip of the manipulator shown in figure 3. This manipulator has two degrees of freedom, a rotation about a vertical axis at the base and a second rotation about a horizontal axis. We move each joint in a range of 180° in 40 equal steps (see figure 8 and figure 9). Note that these two factors of variation are independent, just like all other factors (i.e. we record all possible combinations between the two).

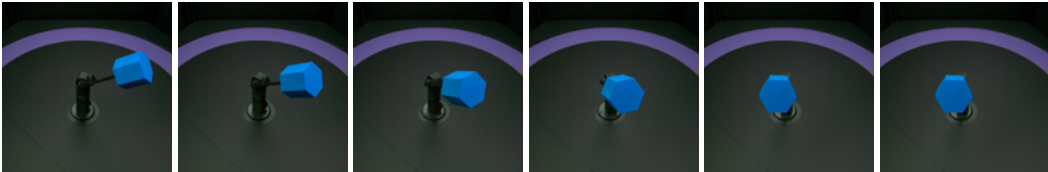

Figure 8: Motion along first DOF while maintaining the other factors constant. Note that in total we record 40 steps, here we only show 6 due to space constraints.

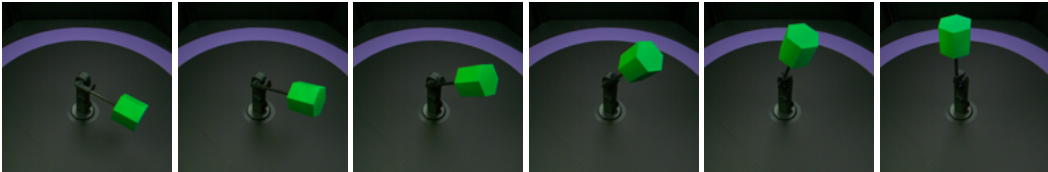

Figure 9: Motion along second DOF while maintaining the other factors constant. Note that in total we record 40 steps, here we only show 6 due to space constraints.

## 3.2 Simulated Data

In addition to the real-world dataset we recorded two simulated datasets of the same setup, hence all factors of variation are identical across the three datasets. One of the simulated datasets is designed to be as realistic as possible and the synthetic images are visually practically indistinguishable from real images (see figure 1 middle). For the second simulated dataset we used a deliberately simplified model (see figure 1 left), which allows to investigate transfer from simplified models to real data.

The synthetic data was generated using Autodesk 3ds Max(2018). Most parts of the scene were imported from SolidWorks CAD files that were originally used to construct the experimental stage including the manipulator and 3D printing of the test objects. The surface shaders are based on Autodesk Physical material with hand-tuned shading parameters, based on full resolution reference images. The camera poses were initialized from the CAD data and then manually fine-tuned using reference images. The realistic synthetic images were obtained using the Autodesk Raytracer (ART) with three rectangular light sources, mimicking the LED panels. The simplified images were rendered with the Quicksilver hardware renderer.

## 4 First Experimental Evaluations of (unsupervised) Disentanglement Methods on Real-World Data

Some fields have been able to narrow the gap between simulation and reality [56, 5, 23], which has led to remarkable achievements (e.g. for in-hand manipulation [1]). In contrast, for disentanglement methods this gap has not been bridged yet, state-of-the-art algorithms seem to have difficulties to transfer learned representations even between toy datasets [38]. The proposed dataset will enable the community to systematically investigate how such transfer of information between simulations with different degrees of realism and real data can be achieved. In the following we present a first experimental study in this directions.

## 4.1 Experimental Protocol

We apply all the disentanglement methods ($\beta$-VAE, FactorVAE, $\beta$-TCVAE, DIP-VAE-I, DIP-VAE-II, AnnealedVAE) which were used in a recent large-scale study [38] to our three datasets. Due to space constraints, the models are abbreviated with numbers one to five in the plots in the same order. We use (`disentanglement_lib`) and we evaluate on the same scores as [38]. In all the experiments, we used images with resolution 64x64. This resolution is used in the recent large-scale evaluations and by state-of-the-art disentanglement learning algorithms [38]. Each of the six methods is trained on each of the three datasets with five different hyperparameter settings (see table 2 in the appendix for details) and with three different random seeds, leading to a total of 270 models. Each model is trained for 300,000 iterations on Tesla V100 gpus. Details about the evaluation metrics can be found in appendix C.

## 4.2 Experimental Results

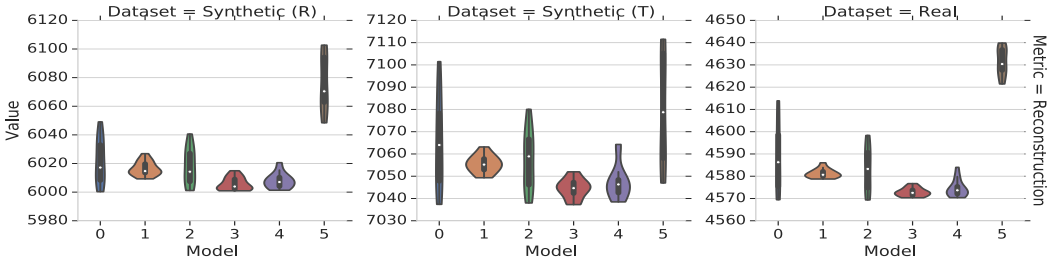

Figure 10: Reconstruction scores of different methods (0=$\beta$-VAE, 1=FactorVAE, 2=$\beta$-TCVAE, 3=DIP-VAE-I, 4=DIP-VAE-II, 5=AnnealedVAE) on the realistic synthetic dataset, the toy synthetic dataset and the real dataset.

**Reconstruction Across Datasets:** Figure 10 shows that there is a difference in reconstruction score across datasets: The score is the lowest on real data, followed by the realistic simulated dataset (R) and the simple toy (T) images. This indicates that there is a significant difference in the distribution of the real data compared to the simulated data, and that it is harder to learn a representation of the real data than of the simulated data. However, the relative behaviour of different methods seems to be similar across all three datasets, which indicates that despite the differences, the simulated data may be useful for model selection.

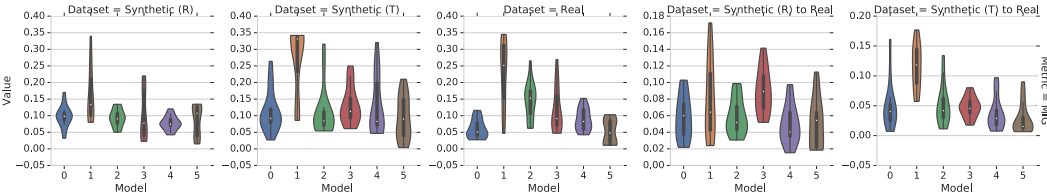

Figure 11: The Mutual Information Gap (MIG) scores attained by different methods for the following evaluations (from left to right): (a) trained and evaluated on synthetic realistic, (b) trained and evaluated on synthetic toy, (c) trained and evaluated on real, (d) trained on synthetic realistic and evaluated on real, (e) trained on synthetic toy and evaluated on real. The variance is due to different hyper-parameters and seeds.

**Direct Transfer of Representations:** In figure 11 we show the Mutual Information Gap (MIG) scores attained by different methods for different evaluations. The same plots for different metrics look qualitatively similar (see figure 22 in the appendix). Given the high variance, it is difficult to make conclusive statements. However, it seems quite clear that all methods perform significantly better when they are trained and evaluated on the same dataset (three plots on the left). Direct transfer of learned representations from simulated to real data (two plots on the right) seems to work rather poorly.

Figure 12: Rank-correlation of the DCI disentanglement scores of different models (including hyperparameters) across different data sets.

**Transfer of Hyperparameters:** We have seen that transferring representations directly from simulated to real data seems to work poorly. However, it may be possible to instead transfer information at a higher-level, such as the choice of the method and its hyperparameters as an inductive bias.

In order to quantitatively evaluate whether such a transfer is possible, we pick the model (including hyperparameters) which performs best in simulation (according to a metric chosen at random), and we compute the probability of outperforming (according to a metric and seed chosen randomly) a model which was chosen at random. If no transfer is possible, we would expect this probability to be $50\%$.

However, we find that model selection from realistic simulated renderings (R) outperforms random model selection $72\%$ of the time while transferring the model from the simpler synthetic images (T) to real-world data even beats random selection $78\%$ of the time.

This finding is confirmed by figure 12, where we show the rank-correlation of the performance of models (including hyperparameters) trained on one dataset with the performance of these models trained on another dataset. The performance of a model trained on some dataset seems to be highly correlated with the performance of that model trained on any other dataset. In figure 12 we use the DCI disentanglement metric as a score, however, qualitatively similar results can be observed using most of the disentanglement metrics (see figure 25 in the appendix).

**Summary** These results indicate that the simulated and the real data distribution have some similarities, and that these similarities can be exploited through model and hyperparameter selection. Surprisingly, it seems that the transfer of models from the synthetic toy dataset may work even better than the transfer from the realistic synthetic dataset.

## 5 Conclusions

Despite the intended applications of disentangled representation learning algorithms to real data in fields such as robotics, healthcare and fair decision making [e.g. 6, 10, 20], state-of-the-art approaches have only been systematically evaluated on synthetic toy datasets. Our work effectively complements related efforts [e.g. 38] to address current challenges of representation learning, offering the possibility of investigating the role of inductive biases, sample complexity, transfer learning and the use of labels using real-world images.

A key aspect of our datasets is that we provide rendered images of increasing complexity for the *same* setup used to capture the real-world recordings. The different recordings offer the possibility of investigating the question if disentangled representations can be transferred from simulation to the real world and how the transferability depends on the degree of *realism* of the simulation. Beyond the evaluation of representation learning algorithms, the proposed dataset can likewise be used for other tasks such as 3D reconstruction and scene rendering [12] or learning compositional visual concepts [21]. Furthermore, we are planning to use the novel experimental setup for recording objects with more complicated shapes and textures under more difficult conditions, such as dependence among different factors.

**Acknowledgments**

This research was partially supported by the Max Planck ETH Center for Learning Systems and Google Cloud. We thank Alexander Neitz and Arash Mehrjou for useful discussions. We would also like to thank Felix Grimminger, Ludovic Righetti, Stefan Schaal, Julian Viereck and Felix Widmaier whose work served as a starting point for the development of the robotic platform in the present paper.

## Footnotes

[3]https://github.com/rr-learning/disentanglement_dataset

[4]https://github.com/deepmind/dsprites-dataset

[5]https://cs.nyu.edu/~ylclab/data/norb-v1.0-small/

[6]http://www.scottreed.info/files/nips2015-analogy-data.tar.gz

[7]https://github.com/deepmind/3dshapes-dataset/

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
