[Supplementary Material]

# A Platform

(a) View from 30° angle from the top, showing the manipulator with a red cube in the middle.

(b) Frontal view without the black shielding, showing the recording from three different camera heights.

(c) View from the top showing the three light centered light panels.

(d) View from the bottom, showing the rotating table (turquoise ring) and the manipulator in the center.

Figure 13: Different rendered images of the recording setup.

Figure 14: The mechanical platform for recording the real-world dataset.

## A.1 Difference between Realistic Simulations and Real-World Images

Figure 15: A technical drawing of the manipulator, dimensions are given in mm.

| (a) Realistically simulated sample. | (b) Real-world sample. |

Figure 16: Comparison of the realistically simulated and the real-world dataset for one example. The detailed procedure and the availability of CAD files used e.g. for the 3D printed robotic arm ensures a close overlap between both examples. Only small details and a high resolution show the differences e.g. the crack in the floor in the left corner which is only visible in the right image or shadings from the light on the object. Both pictures have a resolution of $512 \times 512$.

# B  A Discussion on Disentangled Representations and their Transfer

Empirically, we observed that some of the best trained models were able to disentangle, though imperfectly, the factors of camera height, background colors, object sizes and the motions along the first and second degrees of freedom. Whereas, they performed poorly in disentangling the factors of some object shapes (for e.g. pyramid and cone) and some colors (for e.g. olive and brown). This may well be because of less pixel variation in the respective factors. The features of camera height and background color cause the most difference (maximum L2 distance) in image space. Similarly the object positions at different joint configurations (not the consecutive frames) also have big L2 distance which may explain why the models focus more on learning these factors.

The image reconstructions become more blurry as we move from the simple simulated dataset to the more complex real-world dataset, which can be seen by the reconstruction scores in figure 10. It has been previously noted that the use of overly simplistic priors like standard normal Gaussians in VAE models [25] can lead to overregularization [53]. To achieve disentanglement in VAE models, [18] put higher weights ($\beta$>1) on the KL divergence which decreases the reconstruction quality. With the increased complexity in the dataset, the decrease in reconstruction quality becomes even more pronounced, as illustrated in figure 17, 18 and 19.

In figures 20 and 21, we show the reconstruction results of transfers from simple simulated and realistic simulated datasets to the real-world dataset. The models completely fail in transferring the representations from the simple simulated to the real-world data. On the other hand, in the realistic simulated to real-world transfer (figure 21) the models almost always reconstruct the correct background strip color, manipulator pose and the camera height factor. However, the object properties seem to differ a lot. This shows that in the case of complex environments, the models put more focus on learning the environment to get the better reconstruction accuracy than to learn the important but relatively small changing factors. This result for VAE models has also been confirmed by [16].

Figure 17: Image reconstruction of Factor VAE model on the low quality simulated dataset.

# C  Details of the Experimental Protocol

**Metrics:**  Various methods to validate a learned representation for disentanglement based on known ground truth generative factors $G$ have been proposed [e.g. 11, 47, 7, 24]. This has for example

Figure 18: Image reconstruction of Factor VAE model on the realistic simulated dataset.

Figure 19: Image reconstruction of Factor VAE model on the real dataset.

been expressed as the mutual information of a single latent dimension $Z_i$ with generative factors $G_1, \ldots, G_K$ [47], where in the ideal case each $Z_i$ has some mutual information with one generative factor $G_k$ but none with all the others. Similarly, [11] trained predictors (e.g., Lasso or random forests) for a generative factor $G_k$ based on the representation $\mathbf{Z}$. In a disentangled model, each

Figure 20: Image reconstructions of the Factor VAE model trained on toy dataset and tested on the real-world dataset. All images in the uneven columns are real, and to the right of each real image is its reconstruction.

Figure 21: Image reconstructions of the Factor VAE model trained on realistic simulated dataset and tested on the real-world dataset. All images in the uneven columns are real, and to the right of each real image is its reconstruction.

dimension $Z_i$ is only useful (i.e., has high feature importance) to predict one of those factors. [52] proposed an interventional robustness score. The graphical model of [52] adapted to our setup is illustrated in figure 2. Another form of validation, especially without known generative factors is the visual inspection of "latent traversals" [see e.g. 7].

**Architecture**   All the models used the same convolutional encoder and decoder architecture with the fixed latent size of 10.

| Model | Parameter | Values |
|-------|-----------|--------|
| $\beta$-VAE | $\beta$ | $[1,\ 2,\ 4,\ 6,\ 8]$ |
| AnnealedVAE | $c_{max}$ | $[5,\ 10,\ 25,\ 50,\ 75]$ |
| | iteration threshold | 100000 |
| | $\gamma$ | 1000 |
| FactorVAE | $\gamma$ | $[10,\ 20,\ 30,\ 40,\ 50]$ |
| DIP-VAE-I | $\lambda_{od}$ | $[1,\ 2,\ 5,\ 10,\ 20]$ |
| | $\lambda_d$ | $10\lambda_{od}$ |
| DIP-VAE-II | $\lambda_{od}$ | $[1,\ 2,\ 5,\ 10,\ 20]$ |
| | $\lambda_d$ | $\lambda_{od}$ |
| $\beta$-TCVAE | $\beta$ | $[1,\ 2,\ 4,\ 6,\ 8]$ |

Table 2: Hyperparameters of different methods.

Table 3: Encoder and Decoder architecture for the main experiment.

| Encoder | Decoder |
|---------|---------|
| Input: $64 \times 64 \times$ number of channels | Input: 10 |
| $4 \times 4$ conv, 32 ReLU, stride 2 | FC, 256 ReLU |
| $4 \times 4$ conv, 32 ReLU, stride 2 | FC, $4 \times 4 \times 64$ ReLU |
| $4 \times 4$ conv, 64 ReLU, stride 2 | $4 \times 4$ upconv, 64 ReLU, stride 2 |
| $4 \times 4$ conv, 64 ReLU, stride 2 | $4 \times 4$ upconv, 32 ReLU, stride 2 |
| FC 256, F2 $2 \times 10$ | $4 \times 4$ upconv, 32 ReLU, stride 2 |
| | $4 \times 4$ upconv, number of channels, stride 2 |

**Training Hyperparameters**   The training hyperparameters were kept fixed for each of the considered methods.

Table 4: Common hyperparameters for training.

| Parameter | Values |
|-----------|--------|
| Batch size | 64 |
| Latent space dimension | 10 |
| Optimizer | Adam |
| Adam: beta1 | 0.9 |
| Adam: beta2 | 0.999 |
| Adam: epsilon | 1e-8 |
| Adam: learning rate | 0.0001 |
| Decoder type | Bernoulli |
| Training steps | 300000 |

# D   Detailed Experimental Results

Figure 22: Disentanglement scores attained by different methods for different metrics (row) and evaluations (column, from left to right): (a) trained and evaluated on synthetic realistic, (b) trained and evaluated on synthetic toy, (c) trained and evaluated on real, (d) trained on synthetic realistic and evaluated on real, (e) trained on synthetic toy and evaluated on real. Methods are abbreviated (0=$\beta$-VAE, 1=FactorVAE, 2=$\beta$-TCVAE, 3=DIP-VAE-I, 4=DIP-VAE-II, 5=AnnealedVAE).

Figure 23: Rank correlation of different metrics on different data sets. Overall, we observe that all metrics except for Modularity seem to be at least mildly correlated on all data sets.

Figure 24: Disentanglement scores on real data vs simulated datasets. There seems to be a positive correlation for at least three metrics.

Figure 25: Rank-correlation of different disentanglement metrics across different data sets. Good hyperparameters seem to transfer well between datasets according to at least three metrics.