[Reviews · NeurIPS 2019]

Reviewer 1



ORIGINALITY: although the paper introduces no new methodology, their dataset is to my knowledge a first-of-its-kind and their design, construction, and use of the robotic camera rig to generate is novel and unusual in the field. This kind of work is innovative and shows an attention to experimental design that is sometimes lacking in machine learning. QUALITY: All parts of the paper (dataset design, camera design and construction, dataset construction, and empirical studies) are well executed. I have no complaints at this time. CLARITY: The paper is well-written and fairly easy to understand across all parts. It covers a lot of ground and as such, must sequester a lot of details in the appendix: robot schematics, model hyperparameters, and detailed results, in particular. However, they are nonetheless included and described in detail, enabling reproducibility. SIGNIFICANCE: As disentanglement is not my area of expertise, it is hard to say for sure how the community will respond, but in my experience, the appetite among researchers for clean, detailed, easy-to-use datasets -- especially with ground truth labels -- is enormous, and as such, I expect this dataset will be widely used and will become a de facto standard. I wouldn't be surprised if it picks up a hundred citations within a few years, if not faster. Here is a laundry list of questions for the authors (I don't have a ton): * It seems unfortunate that the "3Dprinted-realistic" dataset is a lower resolution (256 x 256) than the real world data (512 x 512)? Is this a limitation of the creation process using Autodesk (the text indicates that at full res, the differences between real and simulated images are more obvious)... * When transferring from lower resolution, e.g., synthetic, images, is it necessary to downsample the higher resolution, e.g., real images? If so, how might this impact performance?

Reviewer 2



This paper addresses a really important issue in the disentanglement literature, which is the gap between toy datasets and real worlds. As someone who also worked in this area, I think this is indeed something that deserves a lot more attention. I do think using a 3D-dataset that was actually recorded rather than rendered using some software to test disentanglement is a very valuable contribution. Aside from the contributions, this is very well written paper. The authors do a great job at explaining the motivations and the protocol for designing the dataset and experiments. While I extremely appreciate the effort that the authors put in designing this dataset, as well as highlighting the importance of the issues in the current disentanglement literature, I have some concerns about the contribution of the paper, which I appreciate if the authors could comment on. 1) As the authors point out, most of the research on disentanglement have focused on developing new algorithms to achieve it. I think at some point, instead of asking the question “how can we do disentanglement?”, we need to start asking the question “Is disentanglement helpful (for downstream tasks)? ”. It seems like everyone is assuming for sure that a disentangled representations will achieve a better generalization performance on any task, as so almost all papers are just bothered to evaluate the disentanglement itself. A good example of a paper that touched on this point is [1]. From this perspective, what I would have liked to see from a new real-world disentanglement dataset was to have a series of real-world tasks associated with it, and instead of evaluating disentanglement based on the one-to-one correspondence between latent codes and generative factors, we evaluate based on the performance on the real-world tasks (which disentanglement supposedly aids). While the proposed dataset has some interesting new factors compared to previous datasets such as background color and camera-angle, still the main difference to previous datasets seems to be the difficulty of the “image” itself. In real-world, I would like to believe that we are not solely interested in “achieving” disentanglement, but would like to use it to perform well on task associated to some real-world data. I understand this is considered a bit out-of-scope for this paper, but I still appreciate if the authors could comment on it. 2) On page 4, the authors state that [2] shows that training low resolution images results in the instability therefore random seed and hyperparameters being more important than the model. Could the authors clarify this? I’m not sure if the reason for this necessarily has to do with the images being low resolution, but simply just the nature of these datasets (or the methods themselves). 3) What is the architecture used to train the new dataset? 4) Could the authors maybe comment on the difficulty of disentanglement (or reconstruction) for each individual feature? (which features are harder to disentangle?) I would be also interested to see which features cause the most difference in image space. In other words, if I take some data from this dataset and keep all the features the same, but change one feature by 1 unit, what would be the L2 difference between the corresponding images? Refs: [1] Sjoerd van Steenkiste, Francesco Locatello, Jürgen Schmidhuber, and Olivier Bachem. Are disentangled representations helpful for abstract visual reasoning? arXiv preprint arXiv:1905.12506, 2019. [2] Francesco Locatello, Stefan Bauer, Mario Lucic, Sylvain Gelly, Bernhard Schölkopf, and Olivier Bachem. Challenging common assumptions in the unsupervised learning of disentangled representations. arXiv preprint arXiv:1811.12359, 2018

Reviewer 3



This paper addressed the issues of representative learning, and then systematically designed a setup to collect a new set of datasets to help with evaluating/addressing the representative learning issues. The authors then conducted an empirical study of existing approaches on this new dataset with some confirmative conclusions. The main value of this paper lies in the creation of the new dataset.

[Author Response · NeurIPS 2019]

We thank all reviewers for their valuable and constructive feedback. The consensus appears to be that the paper is well written, well motivated and easy to follow (R1, R2). We also appreciate the assessment of our work as *"innovative and unusual in the field"* (R1) and *"a very valuable contribution"* (R2).

**Questions and Comments of Reviewer 1**

- **Resolution of the "3Dprinted-realistic" dataset.** Re-rendering and releasing higher resolution images is an important future work we plan to address. It is however beyond the scope of this paper due to the significant engineering challenges it entails. To illustrate the point, the rendering of 1m high resolution (512x512) images for the '3Dprinted-realistic' image dataset, would roughly take 140 days of continuous computation using our existing infrastructure. Moreover, SOTA disentanglement learning algorithms [37] (Locatello et al., ICML 2019) still use 64x64 resolution, such that already the provided 256x256 resolution is a significant improvement and for a resolution of 512x512 we directly provide the real-world dataset.

- **Downsampling of real-world images for transfer learning.** Thanks for pointing this out. We will make clear in the final version that in all experiments we used images with resolution 64x64. This resolution is used in recent large-scale evaluations and by SOTA disentanglement learning algorithms [37](Locatello et al., ICML 2019).

- **Link to code or the dataset.** Unfortunately, the NeurIPS' policy does not allow to provide any external links in rebuttal documents. However, we guarantee to make all datasets and trained models publicly available before publication.

**Questions and Comments of Reviwer 2**

- ***"Is disentanglement helpful in general? and can authors comment on tasks associated with real-world disentanglement?"*** Thanks for pointing this out. We agree that currently there are multiple research directions for disentangled representation learning. One of them is the suggested investigation of the usefulness of disentangled representations. Our work is encouraging that and allowing to investigate the effectiveness of disentangled representations with access to ground truth labels on real-world data.
Prior work e.g. [20] (Higgins et al. (ICML 2017) already indicated that disentanglement is useful for reinforcement learning. We believe that applying these ideas on the real-world platform is an interesting direction and we will update the future work section.
As pointed out by the reviewer, the effort to find a thorough answer to these questions is significantly beyond the scope of one paper. However, we are already looking into replicating the analyses of the provided reference (van Steenkiste et al., arxiv 2019)[1] and a new fairness approach (Locatello et al. arxiv 2019)[2], both illustrating the benefits of disentanglement, on our real-world data set.

- ***"The authors state that the training on low resolution images results in the instability therefore random seed and hyper-parameters being more important than the model. Could the authors clarify this?"*** Our intention was not to suggest that this instability is necessarily due to the low resolution of those datasets. We agree that this is slightly misleading and we will rephrase this sentence in the final version.

- **Used architecture.**
All the methods use the same convolutional encoder and decoder where the latent dimension is fixed to be 10.
$Encoder : input(64 \times 64 \times 3) \rightarrow 4 \times (4 \times 4conv, 32ReLU, stride2) \rightarrow FC1(256) \rightarrow FC2(10)$
$Decoder : 10 \rightarrow FC1(256), ReLU \rightarrow FC2(4 \times 4 \times 64), ReLU \rightarrow 3 \times (4 \times 4upconv, 64ReLU, stride2) \rightarrow 4 \times 4upconv, 3, stride2$
This can be considered the standard architecture and is used in e.g. [4,11,19,38]. We will make sure to add this information in the final paper.

- ***"Could the authors maybe comment on the difficulty of disentanglement (or reconstruction) for each individual feature? (which features are harder to disentangle?) I would be also interested to see which features cause the most difference in image space."*** Empirically, we observed that some of the best trained models were able to disentangle, though imperfectly, the factors of camera height, background colors, object sizes and the motions along the first and second degrees of freedom. In contrast, they performed poorly in disentangling the factors of some object shapes (for e.g. pyramid and cone) and some colors (for e.g. olive and brown). This may well be because of less pixel variation in the respective factors. The features of camera height and background color cause the most difference (maximum L2 distance) in image space. Similarly the object positions at different joint configurations (not the consecutive frames) also have big L2 distance which may explain why the models focus more on learning these factors. We will add a detailed analysis to the final version.

[1] van Steenkiste, S., Locatello, F., Schmidhuber, J., and Bachem, O. Are disentangled representations helpful for abstract visual reasoning? arXiv preprint arXiv:1905.12506, 2019

[2] Locatello, F., Abbati, G., Rainforth, T., Bauer, S., Schölkopf, B. and Bachem, O. On the Fairness of Disentangled Representations. arXiv preprint arXiv:1905.13662. 2019.

[Meta-Review · NeurIPS 2019]

The paper presents a new benchmark dataset to investigate transfer learning and disentangled representations. The Reviewers agree that this dataset represents a meaningful improvement with respect to previous benchmark (more images, higher resolutions, 3D objects) and it might have great impact on the community.